# Analysis of Molecular Networks in the Cerebellum in Chronic Schizophrenia: Modulation by Early Postnatal Life Stressors in Murine Models

**DOI:** 10.3390/ijms221810076

**Published:** 2021-09-17

**Authors:** América Vera-Montecinos, Ricard Rodríguez-Mias, Karina S. MacDowell, Borja García-Bueno, Álvaro G. Bris, Javier R. Caso, Judit Villén, Belén Ramos

**Affiliations:** 1Psiquiatria Molecular, Institut de Recerca Sant Joan de Déu, Santa Rosa 39–57, 08950 Esplugues de Llobregat, Spain; ajvera@fsjd.org; 2Department of Genome Sciences, School of Medicine, University of Washington, 3720 15th Ave NE, Seattle, WA 98195, USA; rr65@uw.edu (R.R.-M.); jvillen@uw.edu (J.V.); 3Departamento de Farmacología y Toxicología, Facultad de Medicina, Universidad Complutense de Madrid, Instituto de Investigación Sanitaria Hospital 12 de Octubre (Imas12), Instituto Universitario de Investigación en Neuroquímica IUIN-UCM, Avda. Complutense s/n, 28040 Madrid, Spain; karinmac@ucm.es (K.S.M.); bgbueno@med.ucm.es (B.G.-B.); agbris@ucm.es (Á.G.B.); jrcaso@med.ucm.es (J.R.C.); 4Centro de Investigación Biomédica en Red de Salud Mental, CIBERSAM (Biomedical Network Research Center of Mental Health), Institute of Health Carlos III, 28029 Madrid, Spain; 5Parc Sanitari Sant Joan de Déu, Doctor Antoni Pujadas 42, 08830 Sant Boi de Llobregat, Spain; 6Faculty of Medicine, University of Vic—Central University of Catalonia, 08500 Vic, Spain; 7Departamento de Bioquímica i Biología Molecular, Facultat de Medicina, Universitat Autònoma de Barcelona, 08193 Bellaterra, Spain

**Keywords:** proteomics, postmortem brain, pathways, networks, schizophrenia

## Abstract

Despite the growing importance of the cerebellum as a region highly vulnerable to accumulating molecular errors in schizophrenia, limited information is available regarding altered molecular networks with potential therapeutic targets. To identify altered networks, we conducted one-shot liquid chromatography–tandem mass spectrometry in postmortem cerebellar cortex in schizophrenia and healthy individuals followed by bioinformatic analysis (PXD024937 identifier in ProteomeXchange repository). A total of 108 up-regulated proteins were enriched in stress-related proteins, half of which were also enriched in axonal cytoskeletal organization and vesicle-mediated transport. A total of 142 down-regulated proteins showed an enrichment in proteins involved in mitochondrial disease, most of which were also enriched in energy-related biological functions. Network analysis identified a mixed module of mainly axonal-related pathways for up-regulated proteins with a high number of interactions for stress-related proteins. Energy metabolism and neutrophil degranulation modules were found for down-regulated proteins. Further, two double-hit postnatal stress murine models based on maternal deprivation combined with social isolation or chronic restraint stress were used to investigate the most robust candidates of generated networks. CLASP1 from the axonal module in the model of maternal deprivation was combined with social isolation, while YWHAZ was not altered in either model. METTL7A from the degranulation pathway was reduced in both models and was identified as altered also in previous gene expression studies, while NDUFB9 from the energy network was reduced only in the model of maternal deprivation combined with social isolation. This work provides altered stress- and mitochondrial disease-related proteins involved in energy, immune and axonal networks in the cerebellum in schizophrenia as possible novel targets for therapeutic interventions and suggests that METTL7A is a possible relevant altered stress-related protein in this context.

## 1. Introduction

Schizophrenia (SZ) is a complex polygenic psychiatric disorder involving dysregulation of multiples pathways [1] with an estimated prevalence up to 1% in the general population [2] and a high heritability up to 79% [3]. Although the etiology of SZ is not fully understood, several hypotheses have been postulated. The neurodevelopmental hypothesis proposes two critical periods of neurodevelopmental vulnerability, namely early life and adolescence. An environmental double-hit in these phases in a genetically predisposed individual is required for the emergence of the disease [4,5]. Based on this hypothesis, cumulative damage in different molecular pathways required for the early development of the central nervous system could contribute to the failure of axonal assembly connections and normal synaptic transmission, which could remain latent until adolescence [6]. In this phase of life, an additional stressor such as psychosocial stress could impact upon these vulnerable pathologic neural circuits, leading to altered functioning of synaptic responses and the emergence of symptoms [7,8,9]. To study the role that psychosocial stress may play in SZ, different animal models have been developed that include prenatal or perinatal stress. Prenatal models may include different types of stressors such as restraint of movement together with water and/or food deprivation [9] and foot-shocks [10] to the mother. Supporting these animal models are other findings from animal experiments that showed that prenatal exposure to stressors leads to learning deficits [11,12]. Furthermore, a study found that post-weaning social isolation induces altered adult behavior as a result of hyperactivity of the hypothalamic–pituitary–adrenal axis [13]. Another animal model used to understand the origin of SZ is the double-hit model in which two hits are required for the emergence of this disorder, the first hit occurring in the prenatal or perinatal phase and a second during adolescence [14,15]. According to the developmental hypothesis for SZ [5,16], the etiological validity is satisfied with these models as they are based on stressors during developmental phase in the origin. The face and construct validity of these models is demonstrated through several behavioral test analyses that have shown SZ-like symptoms in these models, including positive symptoms (deficits in prepulse inhibition and anxiety), negative symptoms (reduced sociability and novelty preference) and general psychopathological symptoms (innate anxiety-like behavior, increased locomotor activity and impaired working memory). Moreover, these models also display SZ-like neurobiological characteristics, such as biochemical alterations in glutamatergic and monoaminergic pathways, neuroinflammatory reactions and oxidative/nitrosative stress responses [17].There is emerging accumulating evidence of the role of the cerebellum in stress-related psychiatric conditions [18]. In the context of schizophrenia, psychosocial stress mediated by cortisol is found to be related to decreased blood flow in the cerebellum and an impairment of a number of cognitive functions [19]. In schizophrenia patients, higher cumulative stress has been found to be associated with higher connectivity between the hippocampus and the cerebellum [20]. Another study in siblings of schizophrenia patients reported increased connectivity between the executive control network and the cerebellum after stress [21]. Further high childhood trauma has been shown to increase the relationship between cortisol reactivity and cerebellum activity in schizophrenia but not in bipolar disorder [22]. In this context, for correct emotional processing, high cortisol reactivity is necessary. These authors show low cortisol reactivity in schizophrenia subjects, which could be associated with high activation in the left cerebellar lobules IV and V, areas that contribute to recognizing and discriminating emotional faces. Thus, altered cortisol reactivity could generate disruption in cerebellar-dependent emotional processing in schizophrenia subjects. Psychosocial stress has been shown to be associated with altered resting-state functional connectivity of the cerebellum in participants with high levels of schizotypal traits [23], suggesting that the cerebellum is a brain area susceptible to psychosocial stress and highly vulnerable in schizophrenia. The cerebellum integrates input signals from different brain areas and is connected to stress-response regions (e.g., hypothalamus, amygdala) that have been shown to be affected in schizophrenia. The cerebellum has the molecular machinery needed to process stress mediators [18] and is the last region to complete neuronal progenitor division, neuronal migration and pruning of dendritic arborization [24]. This lengthy maturation phase of the cerebellum makes this area highly vulnerable to stress-induced error accumulation during development, which may have a significant impact in SZ. Thus, the cerebellum constitutes an attractive area to study as a possible reservoir of failures in multiple pathways through development and during postnatal life.

Despite the growing importance of the cerebellum as a stress target region highly vulnerable to accumulating molecular stress-induced errors in schizophrenia, only a few recent studies have investigated global mRNA [25] or proteomic alterations in this region [26,27]. The main aim of our study was to identify altered protein networks in the cerebellum with potential therapeutic targets for schizophrenia. To achieve this, we compared the proteomic profile in postmortem lateral cerebellar cortex of individuals with chronic schizophrenia (*n* = 12) and control healthy subjects (*n* = 14) that we obtained using single shot liquid chromatography–tandem mass spectrometry analysis (Figure 1A). Furthermore, we also investigated robustly altered proteins of the identified altered protein networks in two separate double-hit postnatal stress murine models induced by maternal deprivation combined with an additional stressor (social isolation or chronic restraint stress).

## 2. Results

### 2.1. Quantitative Proteomic Analyses in Cerebellum in Chronic Schizophrenia

To identify altered proteins related to SZ, we performed a proteomic analysis of human cerebellar lateral cortex protein extracts from 12 male SZ patients and 14 control individuals matched for gender, age and postmortem delay (PMD). No differences were observed between the SZ and control groups for any demographic or tissue-related variables (Appendix A). Using mass spectrometry, a total of 2578 proteins were quantified. A total of 1474 proteins (57%) were quantified in at least seven individuals per group and used for subsequent bioinformatics analyses (Appendix A); 97.8% of the proteins were identified with two or more peptides and 46.5% with five or more peptides. For the individual proteome signature analyses, we examined the similarity of the individual proteome through a correlation matrix. To assess the similarity between the proteomes of the different individuals, we calculated Pearson correlation coefficients and visualized the results in a correlation matrix (Figure 1B). All correlations were above 0.7 (Figure 1B). To assess the similarity between the SZ and control groups, we calculated the Pearson correlation coefficient of the protein intensity means calculated for each group. This correlation was 0.989, indicating that globally the cerebellar proteomes of SZ and control individuals were highly similar (data not shown). Unsupervised hierarchical clustering analysis of the proteomic profiles was able to segregate controls and SZ samples, with the exception of five controls (#8, #7, #4, #12 and #14) (Figure 1C). We further investigated whether the demographic- and tissue-related features could explain the differential clustering of these controls and observed that although four of these controls segregated together, none of the variables influence this segregation (Appendix A). We identified 250 proteins significantly regulated (16.9%) in SZ with an FDR of <0.1 (Appendix A) (142 proteins down-regulated and 108 up-regulated). None of the regulated proteins showed significant correlation with PMD, pH or chlorpromazine dose (FDR < 0.1) (Appendix A) for any altered protein. A volcano plot was used to categorize proteins as up or down-regulated based on the fold change (log_2_ FC) between SZ and control cases and the corrected *p*-value (–log_10_ (q-value) adjusted to FDR < 0.1). Most of the significant changes were displayed between 0.2 and 0.6 Log2 FC (Figure 1 D). We found that only 56 of the 250 altered proteins had been previously reported in a gene expression study in an iPSC model of SZ [28], and that only 16 altered proteins had been reported in a microarray assay in human cerebellum [25] (Figure 1E, Appendix A). Eighty-six altered proteins in the cerebellum had been previously reported as altered in other brain regions in SZ, and 20 proteins had been reported in two recent cerebellar proteomic studies [26,27] using different designs (sample pooling versus individual sample analyses), labelling of peptides (chemical labelling quantification versus label-free quantification), type of lysates (cellular fractioning versus whole extracts), mass spectrometry instruments (LTQ-Orbitrap XL, Thermofisher or DS-11, DeNovix versus Q-Exactive, Thermofisher) and age of SZ subjects, among others (Appendix A).

### 2.2. Gene Ontology Enrichment Analysis

Of the 108 up-regulated proteins, those related to stress were found to be the most enriched in the disease category analysis, comprising 12% of the total altered proteins (Figure 2A and Table 1). The up-regulated group was enriched in proteins related to structural and signaling biological functions (Figure 2B and Table 1) and to six pathways: vesicle-mediated transport, apoptosis, Rho GTPase effectors, signaling by Rho GTPases, axon guidance and the cell cycle (Figure 2C). Half of the altered proteins in the stress category (SOD1, MYH9, PDIA3, MAPK1, RPS3, YWHAE and YWHAZ) were also found in the group of proteins found to be enriched in a number of biological processes and pathways mainly related to axonal development and functioning (Table 1). Furthermore, our analysis showed that COMT is a protein that was only found in the stress category. COMT was not involved in other enriched categories in the cerebellum.

Of the 142 down-regulated proteins, those involved in mitochondrial diseases were found to be the most enriched in the disease category analysis, also comprising 12% of the total altered proteins (Figure 2A and Table 1). Analysis of the biological processes of the down-regulated proteins revealed significant enrichment of terms related to energy metabolism (Figure 2B and Table 1), and two predominant pathways were found to be enriched: the citric acid (TCA) cycle/respiratory electron transport and neutrophil degranulation (Figure 2C). Most of the altered proteins in the mitochondrial disease category (DLD, SLC25A4, NDUFA4, NDUFB5, NDUFB6, NDUFB9, NDUFS1, NDUFA12, UQCRB, CAPN1 and SL25A12) were also found in the group of proteins found to be enriched in energy-related functions (Table 1). CAPN1 was also found in the group of proteins enriched in the neutrophil degranulation pathway category (Table 1).

For the subsequent analyses, the altered proteins also reported in gene expression and proteomic analyses in the cerebellum and in iPSC model for schizophrenia at mRNA level were used (see Appendix A and Figure 1E). We thus investigated the overlap between the altered proteins of the enriched categories and the previously reported changes in these gene expression and proteomic studies. These analyses revealed that 38 and 8 non-redundant altered proteins of the enriched categories in our proteomic study were previously reported in transcriptomic and proteomic reports, respectively (Table 1).

### 2.3. Network Generation from Enriched Pathways in the Cerebellum

For the enriched up-regulated pathways, the network analysis showed overlapping pathways that were mainly related to axonal development and functioning (Figure 3A). The vesicle-mediated transport pathway showed the highest overlap with other pathways, with 52% overlapping with signaling by Rho GTPases proteins, 35% with the cell cycle, 29% with apoptosis and 23% with axon guidance proteins (Figure 3A). Stress-related altered proteins in this network (MYH9, MAPK1, YWHAE and YWHAZ) belonged to at least three groups of proteins enriched in pathways and showed five to nine interactions with other altered proteins of the network (Figure 3A). Further, 14 proteins from the mixed module have been previously reported in proteomic and gene expression analyses (Figure 3A).

Network analysis of enriched pathways for the down-regulated proteins revealed two distinct modules, one related to energy metabolism and the other consisting of neutrophil degranulation pathways (Figure 3B). The neutrophil degranulation proteins overlapped with secretory granular and secretory lumen pathways. Many mitochondrial disease-related altered proteins also formed part of the energy network (DLD, NDUFA4, NDUFB5, NDUFB6, NDUFB9, NDUFS1, NDUFA12 and UQCRB) and belong to respiratory electron transport with the exception of DLD, which belongs to the TCA cycle (Figure 3B). Moreover, nine proteins from the down-regulated network have been previously reported in proteomic and gene expression analyses (Figure 3B).

We used publicly available immunocytochemistry data to assess the expression of proteins of the up-regulated and down-regulated networks in the different layers of the cerebellum. In the up-regulated mixed module, most of the proteins were expressed at medium to high levels in all cerebellar layers with the exception of ARRB1 and SNX5 from axon guidance, which were only expressed in Purkinje neurons (Figure 3A). No information was available for the stress-related protein MYH9. In the down-regulated energy module, we found that all the proteins were widely distributed throughout the cerebellar layers (Figure 3B). For the neutrophil degranulation module, half of the proteins were expressed in all cerebellar layers, while the rest were expressed in two layers with the exception of ATG7, which was only expressed in Purkinje cells, and CD47, which was only expressed in the granular layer (Figure 3B).

### 2.4. Analysis of Altered Robust Candidates in Double-Hit Stress Murine Models

For each network module, we identified one or two robust protein candidates for further investigation of the influence of early postnatal stress based on the detected LFQ-intensity fold change and coefficient of variation (<0.35) (Figure 4; Appendix A). The selected proteins were YWHAZ (vesicle-mediated transport and stress-related protein) and CLASP1 (axonal guidance) for the mixed module, NDUFB9 for the energy module and METTL7A for the neutrophil degranulation module (Figure 4; Appendix A). METTL7A was also found altered in previous gene expression studies (Table 1).

Two independent double-hit murine stress models were used. The models were maternal deprivation combined with an additional stressor: social isolation (MD/Iso) or chronic restraint stress (MD/RS). Altered behavior was confirmed in these models (Appendix A). CLASP1 was down-regulated in the MD/Iso model but up-regulated in the SZ cohort (Figure 4A and Appendix AA). No significant changes were found in the DM/RS model for CLASP1 (Figure 4A). YWHAZ, which was up-regulated in the SZ cohort and previously related to stress, did not show any change in either of the two SZ murine models (Figure 4B and Appendix A). We observed that NDUFB9 levels were reduced in the human SZ cohort and in the MD/Iso model but not in the MD/RS model (Figure 4C and Appendix A). METTL7A showed a decrease in protein expression levels in the chronic SZ cohort and in both murine models (Figure 4D and Appendix A).

## 3. Discussion

Our study characterized the proteomic alterations for the cerebellar cortex in chronic schizophrenia. To the best of our knowledge, this is the first time in which the individual proteomic signature allows most SZ cases to be segregated from healthy controls in unsupervised analysis, similarly as that achieved in some gene expression studies [25,29].

### 3.1. Up-Regulated Proteins Related to Stress and Axonal Functions

A subset of the up-regulated proteins has been previously related to stress diseases. In this context, psychosocial stress has been shown to be mediated by cortisol, resulting in decreased blood flow in the cerebellum, and to be associated with the altered resting state of this area [19,23]. Other types of stress have also recently been found to alter the connectivity of the cerebellum with the hippocampus and its reactivity to cortisol [20,22]. A number of the proteins in the stress category (MYH9, MAPK1, YWHAE and YWHAZ) also form part of the highly interacting proteins of the network comprised of proteins with multiple roles in vesicle-mediated transport, axon guidance, cell cycle and/or signaling by Rho GTPases. In addition, one member of this upregulated network, CLASP1 from axonal guidance, was found to be regulated by stress in one murine model (see discussion of the role in Section 3.1.2). This protein also forms multiple interactions with other partners of the network, suggesting that maternal deprivation and social isolation could have a broader impact on axonal communication in the cerebellum. Together, this evidence suggests that an increase in proteins that have roles in multiple pathways and that are associated with stress conditions in the cerebellum could mediate the altered cerebellar functioning and its connectivity with other brain regions in schizophrenia. In addition, our analyses identify that catechol O-methyltransferase enzyme (COMT) was only found in the stress disease category. COMT participates in the metabolism of catecholamines such as dopamine and has been associated as a risk gene for SZ [30]. In the context of stress, studies have shown that the genetic variants of COMT could modulate the stress response, increasing or decreasing the release to blood of molecules involved in the stress response [31,32]. In SZ, it has been reported the COMT genetic variant with higher activity on the dopamine degradation in the prefrontal cortex could be related to an impairment in the cognitive functions and working memory [33,34]. In the context of neurodevelopmental disorders, only one study associated an altered activity of COMT in the cerebellum to the impairment of executive function in attention-deficit/hyperactivity disorder [35]. However, we have not found functional studies of COMT in the cerebellum in SZ. Thus, in the context of SZ, altered COMT levels found in the cerebellum could lead to altered stress response and dopaminergic metabolism dysfunction. These findings also support the susceptibility of this brain area to psychosocial stress in schizophrenia.

#### 3.1.1. Vesicle-Mediated Transport

In our proteomic study, we found up-regulated pathways related to vesicle-mediated transport. In neurons, altered transport could limit effectiveness in neuronal communication [36]. Defective synaptic transmission and neurotransmitter release [37] decreases in pre-synaptic vesicle proteins [38,39], and altered levels of proteins involved in synaptic vesicle fusion [40,41] have been associated with SZ. However, in our study we found an increase of this pathway that could be indicative of a compensatory strategy. YWHAZ was found to be up-regulated in our proteomic study and to form part of the group of proteins related to stress. However, we did not observe any significant changes in YWHAZ in these early postnatal stress-based models, suggesting that later stress in adolescence or adult life could be responsible for the altered levels detected in postmortem cerebellum in schizophrenia. Supporting our results, other proteomics studies in different brain regions of SZ subjects have shown altered levels of YWHAZ [42,43]. YWHAZ has a role as an adaptor protein of extracellular vesicles (EVs), which involves the stabilization of vesicles and synapsis [44]. The overexpression of YWHAZ found in our study could increase the formation and release of EVs carrying protein or miRNA to the synapsis, which could be a compensatory mechanism for defective synaptic activity in this disorder in the cerebellum (see Section 3.2).

#### 3.1.2. Axon Guidance

Our proteomic study showed an altered axon guidance pathway in the cerebellum. Defects in neuronal connectivity during development have been proposed as an important cause of the etiopathology of SZ [45,46]. A study based on the SZ-GWAS database found the axon guidance pathway to be altered in this disorder [47]. CLASP1 was a protein from axon guidance down-regulated in schizophrenia and further studied in two murine stress models in this study. Our results showed decreased expression of CLASP1 in the maternal deprivation and social isolation murine models, showing that this combination of early postnatal stress leads to the down-regulation of this protein in the cerebellum. However, the addition to chronic restraint postnatal stress was not sufficient for altering CLASP1 levels. Our proteomic study in SZ subjects showed an up-regulation. This protein participates in neurite outgrowth by binding at microtubules [48]. One possible explanation for the up-regulation of CLASP1 in chronic schizophrenia could be the accumulation of this protein in the neuritic growth cone due to a decrease in energy production required for the assembly of CLASP1 at microtubules.

### 3.2. Down-Regulated Proteins Related to Mitochondrial Disease, Energy Functions and Immune Response

The down-regulated proteins were enriched in proteins with an implication in mitochondrial diseases. Most of the proteins of this category were also found in the group of proteins related to oxidative phosphorylation and TCA in this study. In SZ, altered mitochondrial function has been previously reported [49]. Furthermore, our network analyses showed two modules for the down-regulated network, namely energy metabolism and neutrophil degranulation.

#### 3.2.1. Energy Metabolism Module

We found a down-regulation of proteins involved in the energy production in the cerebellum in schizophrenia. The cerebellum represents 11% of human brain weight [50], and the distribution of the energy it uses varies between different cell types. The more energy-demanding functions in Purkinje cells include the production of action potentials and maintenance of postsynaptic receptors, while granule cells consume more energy to propagate action potentials and support the resting potential [51]. In the context of SZ, studies of cerebellar activity showed decreased blood flow in this area during several tasks, including attention, social cognition, emotion and memory [52,53]. Other studies have also reported reduced cerebellar activity and altered connectivity linked to different stresses [19,20,22,23], suggesting that the reduced cerebellar activity observed in schizophrenia could be influenced by stress life events.

Evidence of mitochondrial dysfunction in SZ includes genetic [54], metabolic [55] and enzymatic dysfunctions [56]; anatomical abnormalities [57]; and disturbed levels of proteins of glycolysis, TCA cycle, mitochondrial function and oxidative stress [58,59,60,61]. These studies are in line with our results showing a down-regulation of the energy network built by TCA cycle/respiratory electron transport and several mitochondrial proteins.

In our study, NDUFB9, the most robust candidate of the energy network involved in respiratory electron transport, was reduced in schizophrenia. It was also observed to be down-regulated in the double-hit postnatal stress murine model that combined maternal deprivation with social isolation but not in combination with chronic restraint stress, suggesting that the down-regulation of respiratory electron chain proteins altered in schizophrenia could be a social stress-induced response.

NDUFB9 is one of the multiple energy proteins that we found widely expressed in the cerebellum with moderate expression in the cerebellar granule layer and Purkinje layer. This protein is involved in the assembly of Complex I. Together with the activity of Complex I, NDUFB9 has been investigated in other brain areas in SZ [62,63]. Thus, the altered expression of NDUFB9 observed in this study could contribute to reducing energy metabolism in cerebellar cells through the disruption of Complex I in respiratory electron transport and could consequently decrease the propagation action potentials among the major neuron cell types of the cerebellar cortex in chronic SZ. Our study suggests that this mechanism could be induced by early postnatal social stress.

#### 3.2.2. Neutrophil Degranulation Module

Our study observed down-regulation of several proteins involved in the various processes of the neutrophil degranulation pathway in cerebellar tissue. Neutrophil degranulation is one of the first defense barriers against infection [64]. An imbalance of the immune system is one of the hypotheses underlying SZ [65]. In our study, METTL7A was observed to be consistently down-regulated in both SZ human samples and in both combinations of early postnatal stress murine models, suggesting that altered METTL7A in chronic schizophrenia could be an early event that is induced by different types of combinations of postnatal stress, and that other members of this network could also be influenced by stress. Further studies are needed to investigate this latter possibility.

METTL7A, a member of the METTL family of methyltransferases, is of interest as it has been poorly studied; indeed, only one study investigated its role in RhoBTB1 signaling in maintaining Golgi integrity [66], a function that could be involved in neutrophil degranulation. In the context of neutrophil degranulation, the altered expression of METTL7A could impair Golgi integrity and contribute to the abnormal formation of secretory granules in neutrophils, altering the first defense barrier of the innate immune response in chronic SZ. Limited information is available about METTL7A deregulation in the context of schizophrenia. In line with our results, previous studies have reported a decrease in the RNA and protein levels on the prefrontal cortex (Brodmann area 46/10) and the anterior cingulate cortex (Brodmann area 24) in SZ subjects, respectively [67,68]. Reduced mRNA levels of METTL7A have also been found in the induced pluripotent stem cells (iPSC) model for schizophrenia [28]. However, another study on the prefrontal cortex (Brodmann area 9) reported an increase in the RNA levels of METTL7A [69]. Thus, METTL7A is a putative relevant candidate altered in different biological substrates in the context of schizophrenia that has been found reduced in the cerebellum in this disorder and modulated by stress in our study. Further studies are needed to understand the role of the reduction of METTL7A in the cerebellum in the context of SZ.

The use of the human postmortem brain constitutes a valuable tool to understand the molecular pathways altered in several psychiatric disorders. However, it has limitations. Firstly, confounding factors such as age, postmortem delay and pH must be carefully explored. In our proteomic study, we did not find any association between these variables and the significantly altered proteins in the cerebellum. Secondly, patients with chronic schizophrenia had long-term and heterogeneous antipsychotics medications. Some candidates from the upregulated proteins are involved in drug to drug interactions, which suggest a possible effect of antipsychotic medications on our findings. Although, no associations were found with a chlorpromazine equivalent dose and altered proteins, the effect of antipsychotic could not be completely rule out. Thirdly, our study only included men, who do not represent a real population of this disorder. Fourthly, our study consisted entirely of elderly individuals due to the type of sample available.

## 4. Methods and Materials

### 4.1. Postmortem Human Brain Tissue

Samples from the cerebella of subjects with chronic schizophrenia (*n* = 12) and healthy controls (*n* = 14) were obtained from the neurologic tissue collection of the Parc Sanitari Sant Joan de Déu Brain Bank (Barcelona, Spain) (30) and the Institute of Neuropathology of the Universitari de Bellvitge Hospital (Barcelona, Spain), respectively. We matched SZ and control groups by gender (only male patients were included), age, postmortem delay (PMD) and pH (Appendix A). The last daily chlorpromazine equivalent dose for the antipsychotic treatment of patients was calculated based on the electronic records of the last drug prescriptions administered up to death, as described previously [70]. Human cerebellar lateral cortex was dissected from coronal slabs stored at −80 °C, extending from the pial surface to white matter only including grey matter. See Appendix A for more details.

### 4.2. Label Free Quantification Proteomic Analysis Mass Spectrometry

Protein extracts were prepared as described previously [71]. Protein concentration was determined by Bradford assay (Bio-Rad, Hercules, CA, USA). Proteomic samples were prepared as described in the Appendix A, using a starting protein amount of 200 µg. Peptide mixtures were analyzed by single-shot liquid chromatography–tandem mass spectrometry (LC–MS/MS) in a Q-Exactive mass spectrometer (Thermofisher Scientific, Waltham, MA, USA) using a top 20 data-dependent acquisition method. Mass spectrometry data were analyzed using MaxQuant. A target–decoy database search strategy was used to guide filtering and estimate false discovery rates (FDRs). Peptides matches were filtered to an FDR of ≤0.01. The mass spectrometry proteomics data were deposited in the ProteomeXchange Consortium via the PRIDE repository with the dataset identifier PXD024937. The minimum required peptide length was seven residues. Label-free quantification (LFQ) was selected for individual protein comparisons between control and SZ groups. Proteins quantified in fewer than 7 samples per group were excluded from the analysis. The normalized LFQ intensity referred to the mean intensity of the controls. A significance value for each quantified protein was calculated using Student’s *t*-test and adjusted for multiple-hypothesis testing using the Benjamini–Hochberg method [72]. FDRs were computed for all significant values, and the FDR threshold was set to 0.1. The quantified proteins were imported into the Perseus software platform (version 1.6.1.3 http://coxdocs.org/doku.php?id=perseus:start#cite, accessed on 22 June 2021) for quality control and further analysis [73]. See Appendix A for more details.

### 4.3. Bioinformatic Analysis

To visually identify significantly altered proteins, we plotted the log_2_ of the fold change of normalized LFQ intensities between schizophrenia and control samples along with the FDR adjusted –log_10_ (q-value). We used the Schizophrenia Database [74] and FunRich Tool v.3.1.3 [75] to compare the identified altered proteins to those previously reported in gene expression studies. To perform the disease, gene ontology (GO) and pathway analyses, we used the Webgestalt-mediated ORA method. Disease terms were obtained from the PharmacoGenetics Knowledge Base (PharmGKB) and Gene List Automatically Derived For You (GLAD4U) databases [76,77]. For the GO analysis, we performed a non-redundant enriched categories analysis. For the pathway analysis, we used the Reactome database, and for the network generation, we used String version 11.0 [78]. To perform the screening on protein localization analysis in different tissue layers in the cerebellum we used the Human Protein Atlas database [79]. The enrichment analyses were set to an FDR of 0.1. We selected 0.1 as a threshold for FDR to detect a broad panel of altered proteins and enriched gene ontology categories to provide a global overview and generate a broader picture of the altered networks.

### 4.4. Stress Murine Models

Three pregnant Wistar rats (Harlan Ibérica, Spain) at gestation day 15 were individually housed in a controlled setting. One of the litters was used as a control group and the other as a double-hit model randomly. After birth, at postnatal day (PD) 9, both litters were exposed to maternal deprivation for 24 h. On PD21, the pups were weaned, and one of the litters was isolated for 5 weeks (PD21-56); the pups were housed individually and denied physical contact with their siblings. After isolation, the pups were regrouped (MD/Iso). The other litter was exposed to restraint stress between PD 72 and 78 for 6 h every day (MD/RS). These conditions represent well-suited murine models for the study of neuropsychiatric dysfunctions [80,81]. Group sample sizes were CT, *n* = 11; MD/Iso, *n* = 9; MD/RS, *n* = 7. The animals were subjected to cervical dislocation. The brain was removed from the skull and the cerebellum was excised and frozen at −80 °C until assayed. Samples were homogenized by sonication in PBS (pH = 7) mixed with a protease inhibitor cocktail (Complete^®^, Roche, Spain). After adjusting protein levels, homogenates of cerebellar tissue were mixed with loading buffer, and 15 µg was loaded into an electrophoresis gel, then blotted onto a nitrocellulose membrane with a semi-dry transfer system (Bio-Rad) and incubated with specific antibodies against (1) methyltransferase-like 7A (METTL7A); (2) NADH dehydrogenase [ubiquinone] 1 beta subcomplex subunit 9 (NDUFB9); (3) CLIP-associating protein 1 (CLASP1); (4) 14-3-3 protein zeta/delta (YWHAZ); and (5) beta-actin (β-actin). See Appendix A for more detailed information.

### 4.5. Statistical Analysis

Normal distribution of variables was determined using the D’Agostino–Pearson test. Demographics and tissue feature of the samples were compared between cases and controls using Student’s *t*-test for parametric quantitative variables and the Mann–Whitney U test for non-parametric variables. Spearman or Pearson correlation analysis was carried out to detect association of our proteomic data with other clinical-, demographic- and tissue-related variables (age, postmortem delay, pH, daily chlorpromazine equivalent dose and duration of illness). For experiments with animal models, we performed outliers analysis using the ROUT method, unpaired Student *t*-tests for parametric variables and Mann–Whitney U tests for non-parametric variables. Statistical analysis was performed with Graph Prism version 7.00 (GraphPad Software, San Diego, CA, USA). The significance level was set to 0.05.

## 5. Conclusions

Altogether, our study provides altered stress- and mitochondrial disease-related proteins involved in energy, immune and axonal networks in the cerebellum in schizophrenia, suggesting that the accumulation of altered events in these networks could lead to a failure in the normal cerebellar functions, impairing synaptic response and the defense mechanisms of this region against external harmful injuries in schizophrenia. Further double hit murine models point out that some of the molecular network alterations observed in schizophrenia could be induced by different combinations of stress exposure during the late postnatal developmental phase of the cerebellum. Thus, these results suggest that the cerebellum is an area vulnerable to accumulate molecular errors induced by stress during early postnatal life in schizophrenia. These findings could provide a panel of possible molecular targets and pathways related to stress and mitochondrial function to explore specific antipsychotic treatments in these murine stress models. These pre-clinical studies could be useful for the development of novel combinations of antipsychotics for schizophrenia in future clinical studies.

## Figures and Tables

**Figure 1 ijms-22-10076-f001:**
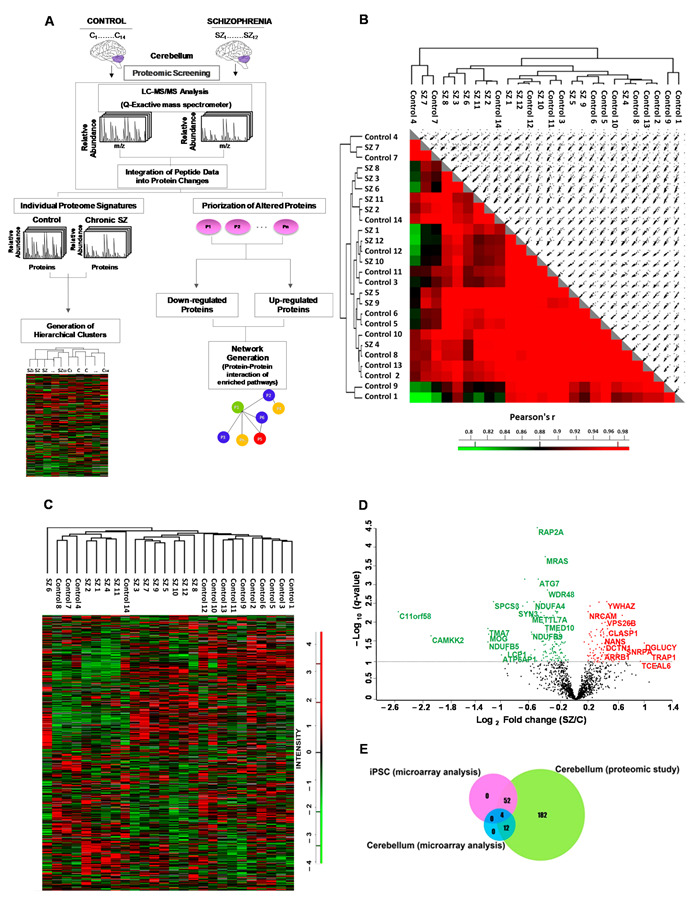
Quantitative proteomic analysis in the cerebellum in chronic schizophrenia. (**A**) Experimental design for proteomic analysis to identify altered pathways in schizophrenia. Protein lysates from the postmortem cerebellum of control C (*n* = 14) and chronic schizophrenia (SZ) patients (*n* = 12) were processed as described. The peptides were separated and analyzed by liquid chromatography (LC) coupled with tandem mass spectrometry. The relative fold change of peptides was integrated into protein changes. The individual protein signatures for each case and control were used to generate hierarchical clusters. The prioritization of altered proteins (P1-Pn represents generic proteins) in SZ was obtained by comparing protein fold changes between control and SZ groups (significant proteins adjusted using an FDR of 0.1). We performed two analyses for these altered proteins in SZ: (i) unsupervised hierarchical clustering analysis generated from quantified proteins in 12 SZ and 14 healthy control samples of postmortem cerebellum; and (ii) generation of networks from significantly enriched pathways by protein–protein interaction. These analyses were performed using Perseus and String, respectively. (**B**) A correlation matrix for 1474 quantified proteins across sample pairs. (**C**) Unsupervised hierarchical clustering analysis was obtained with matrix processing according to the Euclidean distance and z-score aggregation method. Protein profiles were generated from 1474 quantified proteins in 12 SZ and 14 healthy control samples of postmortem cerebellum and were clustered according to the z-score and displayed as a heat map. Green color clusters represent under-expressed proteins. Red color clusters represent overexpressed proteins. (**D**) Volcano plot of the −log_10_ q-value (adjusted *p*-value; FDR (≤0.1)) versus the log_2_ fold change in the cerebellum in SZ relative to healthy control. SZ, schizophrenia; C, control. Up-regulated and down-regulated significant proteins are represented in red and green, respectively. The grey line shows the FDR threshold. (**E**) Venn diagram showing overlap between proteins previously reported in SZ through gene expression analysis obtained from SZDB (human cerebellum and iPSC) and our proteomic study in cerebellum.

**Figure 2 ijms-22-10076-f002:**
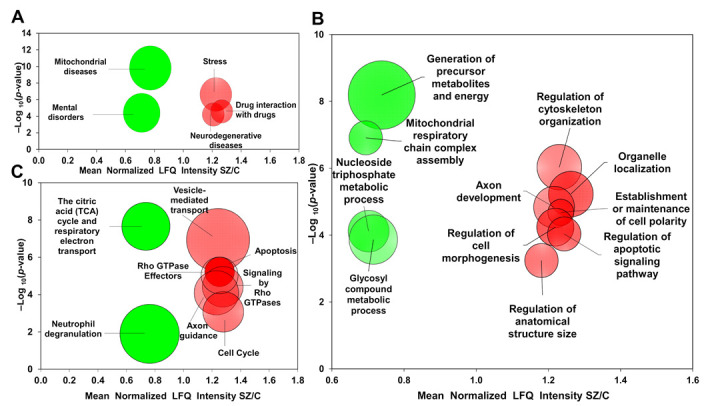
Enrichment analyses from proteome cerebellum in chronic schizophrenia. The bubble chart showing enriched disease categories (**A**), biological processes (**B**) and pathways (**C**) for 142 down-regulated proteins and 108 up-regulated proteins in SZ. (**A**) The enriched categories for the down-regulated proteins were mitochondrial diseases (PA447172) and mental disorders (PA447208), and for the up-regulated proteins they were stress (PA445752), drug interactions with drugs (PA165108622) and neurodegenerative diseases (PA446858). (**B**) Non-redundant enriched biological process categories for down-regulated proteins in SZ were generation of precursor metabolites and energy (GO: 0006091), mitochondrial respiratory chain complex assembly (GO: 0033108), nucleoside triphosphate metabolic process (GO: 0009141) and glycosyl compound metabolic process (GO: 1901657). For up-regulated proteins, the enriched biological functions were regulation of cytoskeleton organization (GO: 0051493), organelle localization (GO: 0051640), axon development (GO: 0061564), establishment or maintenance of cell polarity (GO: 0007163), regulation of cell morphogenesis (GO: 0022604), regulation of apoptotic signaling pathway (GO: 2001233), regulation of anatomical structure size (GO: 0090066) and microtubule-based movement (GO: 0007018). (**C**) The enriched pathway categories in SZ for the down-regulated proteins were citric acid (TCA) cycle/respiratory electron transport (R-HSA-1428517) and neutrophil degranulation (R-HSA-6798695). The enriched pathways for the up-regulated proteins were vesicle-mediated transport (R-HSA-5653656), apoptosis (R-HSA-109581), signaling mediated by Rho GTPase effectors (R-HSA-195258), signaling by Rho GTPases (R-HSA-194315), axon guidance (R-HSA-422475) and cell cycle (R-HSA-1640170). The X-axes show the mean of normalized LFQ intensity in SZ relative to control group for all the proteins that belonged to each category. The Y-axes show the –log_10_ enrichment *p*-value. The bubble size is directly proportional to the number of proteins represented in each enriched category of diseases, biological processes or pathways. Red color represents up-regulated proteins. Green color represents down-regulated proteins. SZ, schizophrenia; C, control.

**Figure 3 ijms-22-10076-f003:**
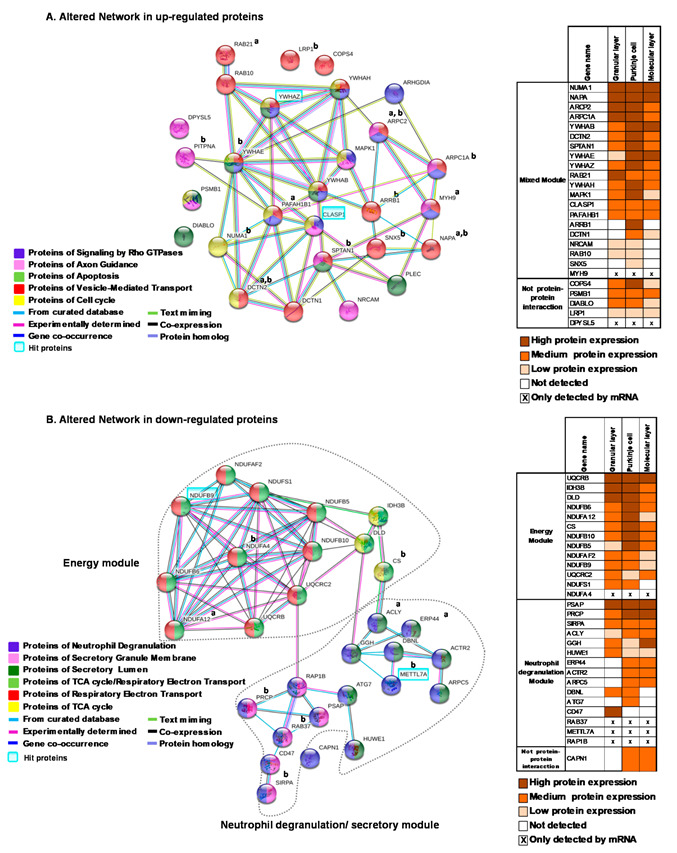
Network generation formed by altered pathways in cerebellum. (**A**) A protein–protein interaction network for up-regulated pathways. (**B**) A protein–protein interaction network for down-regulated pathways. The interaction overview shows how proteins overlap in the different enriched pathways (Figure 2C). Each node represents a protein. Color denotes membership to the module. The colored edge (connections between nodes) represents the type of interaction between nodes. Highlighted gene symbols represent the most robust hit protein for each module. In the mixed module, the candidate selected belonged to at least three different pathways. ^a^ indicates the proteins previously report in the proteomic studies in cerebellum; ^b^ indicates proteins previously reported in the gene expression analysis in iPSC and cerebellum. The right panels show the level of protein expression determined by immunohistochemistry for each protein in the modules and their localization in the different layers in the cerebellum.

**Figure 4 ijms-22-10076-f004:**
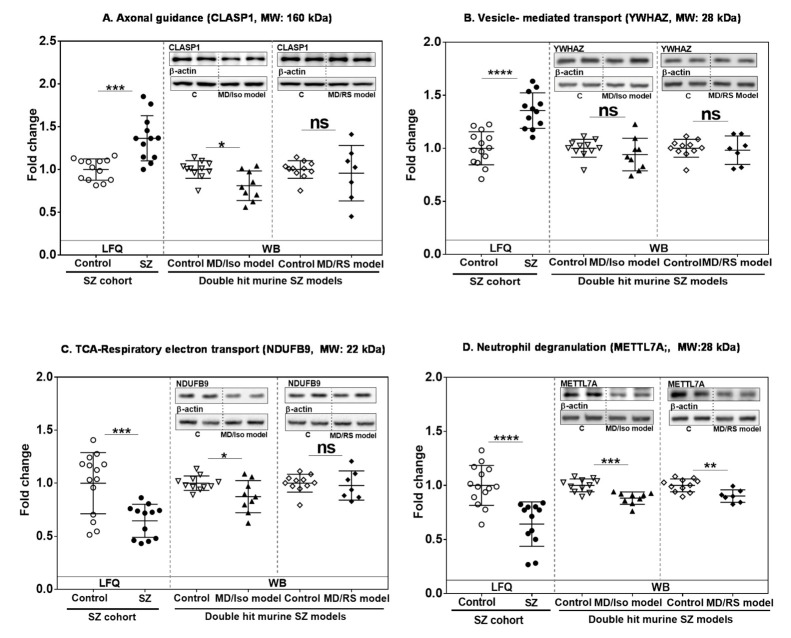
Analysis of hit proteins from altered pathways in a human SZ cohort and two double-hit SZ murine models. Protein levels of candidate hit proteins CLASP1 (**A**), YWHAZ (**B**), NDUFB9 (**C**) and METTL7A (**D**) from the indicated enriched pathway in SZ were analyzed in the cerebellum of the human SZ cohort (control: *n* = 14; SZ = 12) by proteomics, and in the two double-hit SZ murine models, maternal deprivation combined with social isolation (MD/Iso) or chronic restraint stress (MD/RS) (control: *n* = 11; MD/Iso model: *n* = 9; MD/RS model: *n* = 8) by immunoblot. TCA-RET: citric acid cycle-respiratory electron transport. VTM: vesicle-mediated transport. SZ, schizophrenia. LFQ, Label free quantitative. WB, Western blot. Full images of immunoblots are available in Appendix A. Statistical analysis was performed using Student’s *t*-test for samples with normal distribution, and the Mann–Whitney U test was carried out for non-parametric distribution in the MD/RS model for NDUFB9 and the MD/Iso and MD/RS models for CLASP1 and YWHAZ. Protein levels were normalized to the mean of the controls. Individual values represent the protein levels for each subject or animal. Means and standard deviations are shown in the graphs. ns: not significant, * *p* < 0.05, ** *p* < 0.01, *** *p* < 0.001, **** *p* < 0.0001.

**Table 1 ijms-22-10076-t001:** Non-redundant categories of disease, gene ontology and pathways among 250 altered proteins filtered by multiple comparison, FDR = 0.1.

CLASSIFICATION	CATEGORY	PROTEIN OVERLAP IN CATEGORY	Total Number	Observed Number	E	*p-*Value	FDR
	Up-regulated proteins						
**DISEASES**	Stress	SYCRIP; COMT; TXN2; **PDIA3** ^b^; HSPA9; **MYH9** ^a^; NACA ^b^; **MAPK1**; **RPS3**; **SOD1** ^b^; **YWHAE** ^b^; **YWHAZ**; USP7 ^b^	592	13	2.20	2.30 × 10^–7^	6.02 × 10^–4^
	Drug interaction with drug	HSPA9; LRP1 ^b^; ARRB1 ^b^; PPIA; MAPK1; DIABLO; TPT1; YWHAB; YWHAZ	423	9	1.57	2.56 × 10^–5^	3.36 × 10^–2^
	Neurodegenerative diseases	DCTN1; HSPA9; LRP1 ^b^; MAP1B; RTN1 ^b^; SNCG; SNRPD1; SOD1 ^b^; UCHL1	473	9	1.76	6.11 × 10^–5^	5.33 × 10^–2^
**BIOLOGICAL PROCESSES **	Regulation of cytoskeleton organization	ARPC2 ^b^; ARPC1A ^b^; DCTN1; CLASP1; ARHGDIA; LRP1 ^b^; PAFAH1B1 ^a^; **MAPK1**; **RPS3**; SPTAN1 ^b^; PM1; CAPZA2	414	12	2.08	9.29 × 10^–7^	7.08 × 10^–4^
	Organelle localization	DCTN2 ^b^; DCTN1; CLASP1; MAP1B; **MYH9** ^a^; NUMA1 ^b^; PAFAH1B1 ^a^; UCHL1; **YWHAZ**; CADPS; NAPA ^a,b^; LIN7A ^b^	495	12	2.49	5.91 × 10^–6^	2.25 × 10^–3^
	Axon development	RAB10; RAB21 ^a^; ARHGDIA; MAP1B; NRCAM; PAFAH1B1 ^a^; PITPNA ^b^; **MAPK1**; DPYSL5; SPTAN1 ^b^; UCHL1	452	11	2.27	1.46 × 10^–5^	3.71 × 10^–5^
	Establishment or maintenance of cell polarity	RAB10; CLASP1; MAP1B; **MYH9** ^a^; NUMA1 ^b^; PAFAH1B1 ^a^; LIN7A ^b^	168	7	0.85	2.10 × 10^–5^	4.0 × 10^–3^
	Regulation of cell morphogenesis	ARPC2 ^b^; RAB21 ^a^; ARHGDIA; MAP1B; **MYH9** ^a^; NRCAM; PAFAH1B1 ^a^; C1QBP; TPM1; YWHAH	433	10	2.18	5.78 × 10^–5^	6.29 × 10^–3^
	Regulation of apoptotic Signaling pathway	**PDIA3**^b^; HNRNPK; RPS3; **SOD1**^b^; TPT1; **YWHAB**; YWHAE ^b^; **YWHAH**; YWHAZ	369	9	1.86	9.24 × 10^–5^	8.80 × 10^–3^
	Multicellular organismal signaling	NRCAM; ATP2B1; ATP2B3; PAFAH1B1 ^a^; **SOD1** ^b^; **YWHAE** ^b^	472	9	2.38	5.74 × 10^–3^	3.36 × 10^–2^
	Regulation of anatomical structure size	ARPC2 ^b^; ARPC1A ^b^; RAB21 ^a^; MAP1B; NRCAM; PAFAH1B1 ^a^; **SOD1** ^b^; SPTAN1 ^b^; CAPZA2	472	9	2.36	5.74 × 10^–4^	3.36 × 10^–2^
	Cytosolic transport	VPS26B; DCTN1; RAB21 ^a^; SNX5 ^b^; **MAPK1**	138	5	0.69	6.46 × 10^–4^	3.51 v 10^–2^
	Microtubule-basedmovement	DCTN1; RAB21 ^a^; MAP1B; PAFAH1B1 ^a^; **SOD1** ^b^; UCHL1	226	6	1.14	9.50 × 10^–4^	4.83 × 10^–2^
**PATHWAYS**	Vesicle-mediated transport	ARPC2 ^b^; DCTN2 ^b^; ARPC1A ^b^; RAB10; DCTN1; RAB21 ^a^; SNX5 ^b^; LRP1 ^b^; ARRB1 ^b^; PAFAH1B1 ^a^; COPS4; SPTAN1 ^b^; YWHAB; **YWHAE** ^b^; YWHAH; **YWHAZ**; NAPA ^a,b^	670	17	3.84	1.17 × 10^–7^	1.87 × 10^–4^
	Apoptosis	PLEC; DIABLO; PSMB1; SPTAN1 ^b^; YWHAB; **YWHAE** ^b^; YWHAH; **YWHAZ**	174	8	1	6.07 × 10^–3^	1.57 × 10^–3^
	Rho GTPase Effectors	ARPC2 ^b^; ARPC1A ^b^; CLASP1; **MYH9** ^a^; PAFAH1B1 ^a^; **MAPK1**; YWHAB; **YWHAE** ^b^; YWHAH; **YWHAZ**	311	10	1.78	9.32 × 10^–6^	1.86 × 10^–3^
	Signaling by Rho GTPases	ARPC2 ^b^; ARPC1A ^b^; CLASP1; ARHGDIA; **MYH9** ^a^; PAFAH1B1 ^a^; **MAPK1**; YWHAB; **YWHAE** ^b^; YWHAH; **YWHAZ**	446	11	2.56	3.82 × 10^–5^	5.10 × 10^–3^
	Axon guidance	ARPC2 ^b^; ARPC1A ^b^; CLASP1; ARRB1 ^b^; **MYH9** ^a^; NRCAM; PITPNA ^b^; **MAPK1**; PSMB1; DPYSL5; SPTAN1 ^b^; YWHAB	573	12	3.28	8.00 × 10^–5^	8.00 × 10^–3^
	Cell Cycle	DCTN2 ^b^; DCTN1; CLASP1; NUMA1 ^b^; PAFAH1B1 ^a^; **MAPK1**; PSMB1; YWHAB; **YWHAE** ^b^; YWHAH; **YWHAZ**	635	11	3.64	8.40 × 10^–4^	5.60 × 10^–2^
	**Down-regulated proteins**						
**DISEASES**	Mitochondrial diseases	**DLD**; **SLC25A4**^b^; SLC25A5; HSPD1; **ACADVL**^b^; **NDUFA4**^b^; **NDUFB5**; **NDUFB6**; **NDUFB9**; **NDUFS1**; **NDUFA12**^a^; SOD2; **UQCRB**; SLC25A22; **CAPN1**; **SLC25A12**^b^; **NDUFAF2**	353	17	2.32	1.53 × 10^–10^	4.0 × 10^–7^
	Mental disorders	ADH5; GABRA6 ^b^; PCLO ^b^; ANK3 ^b^; MARK1; MOG; ATP1A3 ^b^; PIP4K2A ^b^; SLC17A7; PTPRD ^b^; SMS; SYP ^b^; SYN3 ^b^; PICALM ^b^; SLC25A12 ^b^	679	15	4.45	3.83 × 10^–5^	2.01 × 10^–2^
**BIOLOGICAL** ** PROCESSES**	Generation of precursor metabolites and energy	COX17; CS ^b^; **DLD**; GPD1; **SLC25A4** ^b^; IDH3B; **ACADVL** ^b^; **NDUFA4** ^b^; **NDUFB5**; **NDUFB6**; **NDUFB9**; NDUFB10; **NDUFS1**; **NDUFA12** ^a^; **UQCRB**; UQCRC2; **SLC25A12** ^b^; **NDUFAF2**	365	18	3.36	6.47 × 10^–9^	4.93 × 10^–6^
	Mitochondrial respiratory chain complex assembly	COX17; **NDUFB5**; **NDUFB6**; **NDUFB9**; NDUFB10; **NDUFS1**; **NDUFA12** ^a^; **UQCRB**; NDUFA2	290	11	2.67	7.61 × 10^–5^	9.67 × 10^–3^
	Nucleoside triphosphate metabolic process	**DLD**; GPD1; **NDUFA4**^b^; **NDUFB5**; **NDUFB6**; **NDUFB9**; NDUFB10; **NDUFS1**; **NDUFA12**^a^; **UQCRB**; UQCRC2	421	13	3.88	1.36 × 10^–4^	1.48 × 10^–2^
	Glycosyl compound metabolic process	**DLD**; AHCY ^a^; GPD1; **NDUFA4**^b^; **NDUFB5**; **NDUFB6**; **NDUFB9**; NDUFB10; **NDUFS1**; GMPR2; **NDUFA12**^a^; **UQCRB**; UQCRC2	34	4	3.13	2.59 × 10^–4^	2.45 × 10^–2^
	ATP hydrolysis-coupled transmembrane transport	ATP1A1 ^b^; ATP1A3 ^b^; ATP6V1C1; ATP6AP1 ^b^	37	4	0.34	3.59 × 10^–4^	3.04 × 10^–2^
	Tricarboxylic acid metabolic process	CS ^b^; **DLD**; IDH3B; ACLY ^a^	37	4	0.34	3.59 × 10^–5^	3.04 × 10^–3^
**PATHWAYS**	The citric acid (TCA) cycle and respiratory electron transport	CS ^b^; **DLD**; IDH3B; **NDUFA4** ^b^; **NDUFB5**; **NDUFB6**; **NDUFB9**; NDUFB10; **NDUFS1**; **NDUFA12** ^a^; **UQCRB**; UQCRC2; **NDUFAF2**	171	13	1.79	2.19 × 10^–8^	3.50 × 10^–5^
	Neutrophil degranulation	HUWE1; ARPC5; ACTR2; ATG7; SIRPA ^b^; ERP44 ^a,b^; METTL7A ^b^; DBNL; RAB37 ^b^; ACLY ^a^; PRCP ^b^; PSAP; RAP1B; **CAPN1**; GGH; CD47	487	16	5.07	4.0 × 10^–5^	1.28 × 10^–2^

Total number: number of reference proteins in category/pathways; observed number: proteins in the data set and also in category/pathways; E: expected in the category and adjusted *p*-value is corrected for test multiple. Bold shows proteins from stress or mitochondrial diseases also found in other categories. ^a,b^ shows proteins previously reported in protein level analysis (^a^) and gene expression analyses (^b^) in the cerebellum (Appendix A).

## Data Availability

The mass spectrometry proteomics data have been deposited in the ProteomeXchange Consortium via the PRIDE repository with the dataset identifier PXD024937.

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
