# Peer review of "Analysis of Molecular Networks in the Cerebellum in Chronic Schizophrenia: Modulation by Early Postnatal Life Stressors in Murine Models"

_ijms, 2021, doi:10.3390/ijms221810076_

Round 1

Reviewer 1 Report

              The authors mention protein analysis in cerebellum of patient with chronic schizophrenia and added stress murine models about several proteins.

              These technics in cerebellum are similar to the authors’ previous study Plos one 2020 and several proteins including CALM2 and PRVA enriched in Scz patient. However, these proteins are not included in this study. This study is planning cascade analysis. What is the definition of protein function? Protein is related to several functions and diseases at same time. For example, COMT is related to stress mental disorders, and much more the authors are sorted. Why was FDR set to 0.1?

              It may be difficult to decide this paper’ worth for upper reasons which should be clarified.

Reviewer 2 Report

The manuscript presents a proteomic investigation of cortex samples from scz and healthy control patients followed by a murine model validation follow-up. The manuscript does a good job of presenting a lot of data with good use of supplementary data. A few points for consideration:

-The abstract is confusing in terms of what your methods and results are. It jumps from the postmorterm cortex samples to the double-hit model and explaining results without explaining which results are from which model. The results does a good job of clarifying your approach with the sub-headings and it makes sense within that section. Please re-word the abstract to increase clarity.

-line 126. Please define PMD when it first arises in the text.

-Thank you for including medication information for the post-mortem samples. I see in your statistical approach that you planned to perform correlations with CPZE dose but I do not see this mentioned in the results or supplementary data. This would be interesting given that one of your ontological pathways was drug-interaction with drug.

-why was your FDR set at 0.1 versus 0.05? I think this is acceptable but brief few words for rationale in the methods could be useful.

-Perhaps a future direction could also be suggested for the murine models to look at specific antipsychotic treatment on your identified proteins/pathways to look at response

Reviewer 3 Report

The authors submitted an interesting and translational (across species) study about proteome changes possibly related to schizophrenia and a schizophrenia-like phenotype in mice. However, I would like to point at some issues, which should be addressed with a revision:

  • Line 71, Introduction, "SZ-like symptoms are reported in these models". The authors should add information about how these SZ-like symptoms can be seen in an animal model (construct validity, face validity of an animal model).
  • Lines 79/80, Introduction: The authors should explain in more detail, what the relationship between "cortisol reactivity and cerebellum activity" means in the referenced publications.
  • Lines 136-139, Results: The authors should add a reference for the "previously reported [...] gene expression study". Also, the authors did not use this information, that only 50 of 250 altered proteins were also regulated in gene expression studies, for their subsequent analyses.
  • Lines 256-258, Results: The proteins for further investigation were identified based on fold change and coefficient of variation. Why did the authors not consider previous data on gene expression data?
  • Lines 264-265, Results: The authors stated that the animal models were verified by an altered behavioral phenotype (Supplemental Figure A2). These data and the figure is missing.
  • Lines 271-272, Results: METTL7A seems to be the main finding, because it was wound in the post mortem analysis and could be validated in both animal models. The abstract and the paper should focus more on this key finding.

Round 2

Reviewer 3 Report

The authors significantly improved the manuscript.